# GlanceNets: Interpretabile, Leak-proof Concept-based Models

## Abstract

There is growing interest in concept-based models (CBMs) that combine high-performance and interpretability by acquiring and reasoning with a vocabulary of high-level concepts. A key requirement is that the concepts be *interpretable*. Existing CBMs tackle this desideratum using a variety of heuristics based on unclear notions of interpretability, and fail to acquire concepts with the intended semantics. We address this by providing a clear definition of interpretability in terms of alignment between the model's representation and an underlying data generation process, and introduce GlanceNets, a new CBM that exploits techniques from disentangled representation learning and open-set recognition to achieve alignment, thus improving the interpretability of the learned concepts. We show that GlanceNets, paired with concept-level supervision, achieve better alignment than state-of-the-art approaches while preventing spurious information from unintendedly leaking into the learned concepts.

## 1 INTRODUCTION

Concept-based models (CBMs) are an increasingly popular family of classifiers that combine the transparency of white-box models with the flexibility and accuracy of regular neural nets [Alvarez-Melis and Jaakkola, 2018, Li et al., 2018, Chen et al., 2019, Losch et al., 2019, Chen et al., 2020]. At their core, all CBMs acquire a vocabulary of concepts capturing high-level, task-relevant properties of the data, and use it to compute predictions and produce faithful explanations of their decisions [Rudin, 2019].

The central issue in CBMs is how to ensure that the concepts are *semantically meaningful* and *interpretable* for (sufficiently expert and motivated) human stakeholders. Current approaches struggle with this. One reason is that the notion of interpretability is notoriously challenging to pin down, and therefore existing CBMs rely on different heuristics—such as encouraging the concepts to be sparse [Alvarez-Melis and Jaakkola, 2018], orthonormal to each other [Chen et al., 2020], or match the contents of concrete examples [Chen et al., 2019]—with unclear properties and incompatible goals. A second, equally important issue is *concept leakage*, whereby the learned concepts end up encoding spurious information about unrelated aspects of the data, making it hard to assign them clear semantics [Mahinpei et al., 2021]. Notably, even concept-level supervision is insufficient to prevent leakage [Margeloiu et al., 2021], cf. Fig. 3.

Prompted by these observations, we define interpretability in terms of *alignment*: learned concepts are interpretable if they can be mapped to a (partially) interpretable data generation process using a transformation that preserves semantics. This is sufficient to unveil limitations in existing strategies, build an explicit link between interpretability and disentangled representations, and provide a clear and actionable perspective on concept leakage. Building on our analysis, we also introduce GlanceNets (aliGned LeAk-proof coNCEptual Networks), a novel class of CBMs that combine techniques from *disentangled representation learning* [Schölkopf et al., 2021] and *open-set recognition* [Scheirer et al., 2012] to actively pursue alignment – and guarantee it under suitable assumptions – and avoid concept leakage.

**Contributions:** Summarizing, we: (*i*) Provide a definition of interpretability as alignment that facilitates tapping into ideas from disentangled representation learning; (*ii*) Show that concept leakage can be viewed from the perspective of out-of-distribution generalization; (*iii*) Introduce GlanceNets, a novel class of CBMs that acquire interpretable representations and are robust to concept leakage; (*iv*) Present an extensive empirical evaluation showing that GlanceNets are as accurate as state-of-the-art CBMs while attaining better interpretability and avoiding leakage.

*Submitted to the 38th Conference on Uncertainty in Artificial Intelligence* (UAI 2022). **To be used for reviewing only**.

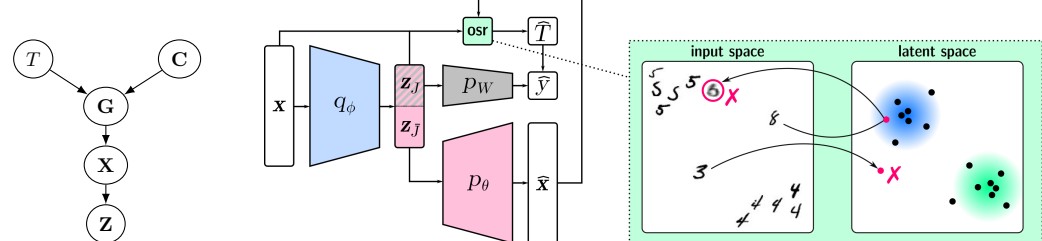

Figure 1: **Left**: The data generation process. **Center:** Architecture of GlanceNets showing the encoder $q_\phi$, decoder $p_\theta$, classifier $p_W$, and open-set recognition step. **Right**: GlanceNets prevent leakage by identifying and rejecting open-set inputs using a combined strategy, shown here for a model trained on digits "4" and "5" only: the "3" is rejected as its embedding falls far away from classes prototypes (colored blobs), while the "8" is rejected as its reconstruction loss is too large.

## 2 CONCEPT-BASED MODELS

Concept-based models (CBMs) comprise two key elements: (i) A learned vocabulary of $k$ high-level concepts meant to enable communication with human stakeholders [Kambhampati et al., 2022], and (ii) a simulatable [Lipton, 2018] classifier whose predictions depend solely on those concepts. Formally, a CBM $f : \mathbb{R}^d \to [c]$, with $[c] := \{1, \ldots, c\}$, maps instances $\mathbf{x}$ to labels $y$ by measuring how much each concept activates on the input, obtaining an activation vector $\mathbf{z}(\mathbf{x}) := (z_1(\mathbf{x}), \ldots, z_k(\mathbf{x})) \in \mathbb{R}^k$, aggregating the activations into per-class scores $s_y(\mathbf{x})$ using a linear map [Alvarez-Melis and Jaakkola, 2018, Chen et al., 2019, 2020], and then passing these through a softmax, i.e.,

$$s_y(\mathbf{x}) := \sum_j w_{yj} z_j(\mathbf{x}), \ \ p(y \mid \mathbf{x}) := \mathrm{softmax}(\mathbf{s}(\mathbf{x}))_y. \tag{1}$$

Each weight $w_{yj} \in \mathbb{R}$ encodes the relevance of concept $z_j$ for class $y$. The activations themselves are computed in a black-box manner, often leveraging pre-trained embedding layers, but learned so as to capture interpretable aspects of the data using a variety of heuristics, discussed below.

Now, *as long as the concepts are interpretable*, it is straightforward to extract human understandable local explanations disclosing how different concepts contributed to any given decision $(\mathbf{x}, y)$ by looking at the concept activations and their associated weights, thus abstracting away the underlying computations. This yields explanations of the form $\{(w_{yj}, z_j(\mathbf{x})) : j \in [k]\}$ that can be readily summarized[1] and visualized [Hase and Bansal, 2020, Guidotti et al., 2018]. Importantly, the score of class $y$ is conditionally independent from the input $\mathbf{x}$ given the corresponding explanation, i.e., $s_y(\mathbf{x}) \perp\!\!\!\perp \mathbf{x} \mid \mathcal{E}(\mathbf{x}, y)$, ensuring that the latter is faithful to the model scores. GlanceNets inherit all of these features.

**Heuristics for interpretability.** Crucially, CBMs are only interpretable insofar as their concepts are. Existing approaches implement special mechanisms to this effect, often pairing a traditional classification loss (such as the cross-

entropy loss) with an auxiliary regularization term [Alvarez-Melis and Jaakkola, 2018, Chen et al., 2019, 2020].

We are interested in particular to variants of concept bottleneck models (CBNMs) [Koh et al., 2020, Losch et al., 2019], which align the concepts using concept-level supervision, possibly obtained from a separate source, like ImageNet [Deng et al., 2009]. From a statistical perspective, this seems perfectly sensible: if the supervision is unbiased and comes in sufficient quantity, and the model has enough capacity, this strategy *appears* to guarantee the learned and ground-truth concepts to match.

**Concept leakage in concept-bottleneck models.** Unfortunately, concept-level supervision is *not* sufficient to guarantee interpretability. [Mahinpei et al., 2021] have demonstrated through simple examples that concepts acquired by CBNMs pick up spurious properties of the data. This phenomenon is known as *concept leakage*.

Intuitively, leakage occurs because in CBNMs the concepts end up unintentionally capturing distributional information about unobserved aspects of the input, failing to provide well-defined semantics. However, a clear definition of leakage is missing, and so are strategies to prevent it: a key contribution of our paper is showing that leakage can be understood from the perspective of domain shift and dealt with using open-set recognition [Scheirer et al., 2012].

## 3 INTERPRETABILITY AND LEAKAGE

The main issue with heuristics used by CBMs is that they are based on unclear notions of interpretability. In order to develop effective algorithms, we propose to view interpretability as a form of *alignment* between the machine's representation and that of its user. This enables us to identify conditions under which interpretability can be achieved, build links to well-understood properties of representations, and leverage state-of-the-art learning strategies.

**Interpretability.** We henceforth focus on the (rather general) generative process shown in Fig. 1 the observations

---

[1]For instance, by pruning those concepts that have little effect on the outcome to simplify the presentation.

$\mathbf{X} \in \mathbb{R}^d$ are caused by $n$ generative factors $\mathbf{G} \in \mathbb{R}^n$, themselves caused by a set of confounds $\mathbf{C}$ (including the label $Y$ [Schölkopf et al., 2012]). Notice that the generative factors *can* be statistically dependent due to the confounds $\mathbf{C}$, but as noted by [Suter et al., 2019], the total causal effect Peters et al. [2017] between $G_i$ and $G_j$ is zero for all $i \neq j$. The generative factors capture all information necessary to determine the observation [Reddy et al., 2022], so the goal is to learn concepts $\mathbf{Z} \in \mathbb{R}^k$ that recover them. The variable $T$ will be introduced later on.

We posit that a (learned) representation is only interpretable if it supports *symbolic communication* between the model and the user, in the sense that it shares the same (or similar enough) semantics to the user's representation. The latter is however generally unobserved. Then, we make a second, critical assumption that *some* of the generative factors $\mathbf{G}_I \subseteq \mathbf{G}$ are interpretable to the user, i.e., they can be used as a proxy for the user's internal representation. Naturally, not all generative factors are interpretable [Gabbay et al., 2021], but in many applications some of them are, e.g., the hair color or noise size in CelebA [Liu et al., 2015].

**Interpretability as alignment.** Under this assumption, if the variables $\mathbf{Z}_J \subseteq \mathbf{Z}$ are *aligned* to the generative factors $\mathbf{G}_I$ by a map $\alpha : \mathbf{g} \mapsto \mathbf{z}_J$ that preserves semantics, they are themselves interpretable. One desirable property is that $\alpha$ does not "mix" multiple $G$'s into a single $Z$. This property can be formalized in terms of *disentanglement* [Eastwood and Williams, 2018, Suter et al., 2019, Schölkopf et al., 2021]. This is however insufficient: we wish the map between $G_i$ and its associated factor $Z_j$ to be "simple", so as to *conservatively* guarantee that it preserves semantics. This makes alignment strictly stronger than disentanglement.

Motivated by this, we say that $\mathbf{Z}_J$ is *aligned* to $\mathbf{G}_I$ if: **(i)** there exists an injective map between indices $\pi : [n_I] \to [k]$, where $[n_I]$ identifies the subset of generative factors indexes in $\mathbf{G}_I$, such that, for all $i, i' \in [n_I]$, $i \neq i'$, and $j = \pi(i)$, it holds that fixing $G_i$ is enough to fix $Z_j$ regardless of the value taken by the other generative factors $G_{i'}$, and **(ii)** the map $\alpha$ can be written as $\alpha(\mathbf{g}) = (\mu_1(g_{\pi(1)}), \ldots, \mu_n(g_{\pi(n_I)}))$, where the $\mu_i$'s are monotone functions. This holds, for instance, for linear transformations of the form $A(g_{\pi(1)}, \ldots, g_{\pi(n_I)})$, where $A \in \mathbb{R}^{n_I \times k}$ is a matrix with no non-zero off-diagonal entries. This second requirement can be relaxed depending on the application.

**Measuring alignment with DCI.** Disentanglement can be measured in a number of ways [Zaidi et al., 2020], but most of them provide little information about how simple the map $\alpha$ is. In order to estimate alignment, we repurpose DCI, a measure of disentanglement introduced by Eastwood and Williams [2018], by fitting a linear model from $\mathbf{z}_J$ to $\mathbf{g}_I$. Further details are included in the Supplementary Material.

**Achieving alignment with concept-level supervision.** It has been shown that disentanglement cannot be achieved in the purely unsupervised setting [Locatello et al., 2019]. This immediately entails that alignment is also impossible in that setting, highlighting a core limitation of [Alvarez-Melis and Jaakkola, 2018]. However, disentanglement can be attained if supervision about the generative factors is available, even only for a small percentage of the examples [Locatello et al., 2020]. As a matter of fact, supervision is used in representation learning to achieve *identifiability*, a stronger condition than – and that entails both of – disentanglement *and* alignment [Khemakhem et al., 2020]. Thus, following CBNMs, we seek alignment by leveraging concept-level supervision.

**Interpretability and concept leakage.** Intuitively, concept leakage occurs when a model is trained on a data set on which (*i*) some generative factors $\mathbf{G}_V \subset \mathbf{G}$ vary, while the others $\mathbf{G}_F = \mathbf{G} \setminus \mathbf{G}_V$ are fixed, and (*ii*) the two groups of factors are statistically dependent. For instance, in the even vs. odd experiment of [Mahinpei et al., 2021], no training examples are annotated with concepts besides 4 and 5. CBNMs with access to supervision on $\mathbf{G}_V$ tend to acquire a latent representation that approximates these factors, and that because of (*ii*) correlates with the fixed factors $\mathbf{G}_F$.

In contrast with previous assessments [Mahinpei et al., 2021, Margeloiu et al., 2021], we notice that point (*i*) can be viewed as a special form of domain shift: the training examples are sampled from a ground-truth distribution $p(\mathbf{X}, \mathbf{G} \mid T = 1)$ in which $\mathbf{G}_F$ is approximately fixed, e.g., $p(\mathbf{G}_F \mid T = 1) = \delta(\mathbf{g}'_F)$ for some vector $\mathbf{g}'_F$, and the test set from a different distribution $p(\mathbf{X}, \mathbf{G} \mid T = 0)$ in which $\mathbf{G}_F$ is no longer fixed. Here, $T$ is a random variable that selects between training and test distribution, see Fig. 3. Since regular CBMs have no strategy to cope with domain shift, they fail to adapt when this occurs.

Motivated by this, we propose then to tackle concept leakage by designing a CBM specifically equipped with strategies for detecting instances that do not belong to the training distribution using open-set recognition [Scheirer et al., 2012]. By estimating the value of the variable $T$ at inference time, we are essentially predicting whether an input was sampled from a distribution similar enough to the training distribution, and therefore can be handled by a model learned on this distribution, or not. This strategy proves very effective in practice, as shown by our empirical evaluation (Section 5.2).

# 4 GLANCENETS

GlanceNets combine a VAE-like architecture [Kingma and Welling, 2014, Rezende et al., 2014] for learning disentangled concepts with a prior and classifier designed for open-set prediction [Sun et al., 2020]. In order to accommodate for non-interpretable factors, the latent representation of GlanceNets $\mathbf{Z}$ is split into two: (i) $k$ concepts $\mathbf{Z}_J$, aligned to the *interpretable* generative factors $\mathbf{G}_I$, that are used for

prediction, and (ii) $\bar{k}$ *opaque* factors $\mathbf{Z}_{\bar{J}}$ that are only used for reconstruction. Specifically, a GlanceNet comprises an encoder $q_\phi(\mathbf{Z} \mid \mathbf{X})$ and a decoder $p_\theta(\mathbf{X} \mid \mathbf{Z})$, both parameterized by deep neural networks, as well as a classifier $p_W(Y \mid \mathbf{Z}_J)$ feeding off the interpretable concepts only. Following other CBMs, the classifier is implemented using a dense layer with parameters $W \in \mathbb{R}^{v \times k}$ followed by a softmax activation, i.e., $p_W(Y \mid \mathbf{z}_J) := \mathrm{softmax}(W\mathbf{z}_J)$, and the most likely label is used for prediction. The overall architecture is shown in Fig. 1.

In contrast to regular VAEs, GlanceNets associate each class to a prototype in latent space through the prior $p(\mathbf{Z} \mid \mathbf{Y})$, which is conditioned on the class and modelled as a *mixture of gaussians* with one component per class. The encoder, decoder, and prior are fit on data so as to maximize the evidence lower bound, defined as [Kingma and Welling, 2019] $\mathbb{E}_{p_D(\mathbf{x},y)}[\mathcal{L}(\theta, \mathbf{x}, y; \beta)]$ with:

$$\mathcal{L}(\theta, \mathbf{x}, y; \beta) := \mathbb{E}_{q_\phi(\mathbf{z}|\mathbf{x})}[\log p_\theta(\mathbf{x} \mid \mathbf{z}) + \log p_W(y \mid \mathbf{z}_J)]$$
$$- \beta \cdot \mathsf{KL}(q_\phi(\mathbf{z} \mid \mathbf{x}) \| p(\mathbf{z} \mid y)) \qquad (2)$$

Here, $p_D(\mathbf{x}, y)$ is the empirical distribution of the training set $D = \{(\mathbf{x}_i, y_i) : i = 1, \dots, m\}$. The first term of Eq. (2) is the likelihood of an example after passing it through the encoder distribution.

The second term penalizes the latent vectors based on how much their distribution differs from the prior and encourages disentanglement. As mentioned in Section 3, learning disentangled representations is impossible in the unsupervised i.i.d. setting [Locatello et al., 2019]. Following [Locatello et al., 2020], and similarly to CBNMs, we assume access to a (possibly separate) data set $\widetilde{D} = \{(\mathbf{x}_\ell, \mathbf{g}_{I,\ell})\}$ containing supervision about the *interpretable* generative factors $\mathbf{G}_I$ and integrate it into the ELBO by replacing the per-example loss $\mathcal{L}$ in Eq. (2) with:

$$\mathcal{L}(\theta, \mathbf{x}, y; \beta) + \gamma \cdot \mathbb{E}_{p_{\tilde{D}}(\mathbf{x},\mathbf{g})} \mathbb{E}_{q_\phi(\mathbf{z}|\mathbf{x})}[\Omega(\mathbf{z}, \mathbf{g})] \qquad (3)$$

where $\gamma > 0$ controls the strength of the concept-level supervision. Following Locatello et al. [2020], the term $\Omega(\mathbf{z}, \mathbf{g})$ penalizes encodings sampled from $q_\phi(\mathbf{z} \mid \mathbf{x})$ for differing from the annotation $\mathbf{g}$. We implement this term using the average cross-entropy loss $\Omega(\mathbf{z}, \mathbf{g}) := -\sum_k g_k \log \sigma(z_k) + (1 - g_k) \log(1 - \sigma(z_k))$, where the annotations $g_k$ are rescaled to lie in $[0, 1]$ and $\sigma$ is the sigmoid.

In order to tackle concept leakage, GlanceNets integrate the open-set recognition strategy of [Sun et al., 2020]. This strategy identifies out-of-class inputs by considering the class prototype $\mu_y := \mathbb{E}_{p(\mathbf{z}|y)}[\mathbf{z}]$ in $\mathbb{R}^k$ defined by the prior distribution and the decoder $p_\theta(\mathbf{x}|\mathbf{z})$. During training, GlanceNets use the training data to estimate: (*i*) a distance threshold $\eta_y$, which defines a spherical subset in the latent space $\mathcal{Z}_y = \{\mathbf{z} : \|\mu_y - \mathbf{z}\| < \eta_y\}$ centered around the prototype of class $y$, and (*ii*) a maximum threshold on the reconstruction error $\eta_{thr}$. If new data points have reconstruction error above $\eta_{thr}$ or they do not belong to any subset

$\mathcal{Z}_y$, they are inferred as open-set instances, i.e., $\hat{T} = 0$. This procedure is illustrated in Fig. 1.

**Remark.** Other disentanglement strategies can be naturally included in GlanceNets to increase its alignment, e.g., with various methods in [Esmaeili et al., 2019], or JL1-VAEs [Rhodes and Lee, 2021]. Since our experiments already show substantial benefits for GlanceNets building on $\beta$-VAEs, we leave these extensions to future work.

# 5 EMPIRICAL EVALUATION

In this section, we present results on several tasks showing that GlanceNets outperform CBNMs [Chen et al., 2020] in terms of alignment and robustness to leakage, while achieving comparable prediction accuracy. The details on the hardware, architectures and hyperparameters are collected in the Supplementary Material.

## 5.1 EVALUATING ALIGNMENT

In a first experiment, we compared GlanceNets with CB-NMs on three classification tasks for which supervision on the generative factors is available. In order to evaluate the impact of this supervision on the different competitors, we varied the amount of training examples annotated with it from $1\%$ to $100\%$. For each increment, we measured prediction performance using accuracy, alignment and explicitness using the lasso variant of DCI.

**Data sets.** We carried out our evaluation on two data sets taken from the disentanglement literature *dSprites* [Matthey et al., 2017] and *MPI3D* [Gondal et al., 2019], and a very challenging real world dataset, *CelebA-64* [Liu et al., 2015]. They all consist in $64 \times 64$ annotated images, with only one channel for dSprites and three for the others. For CelebA, since we are interested in measuring alignment, we considered only those 10 binary generative factors that CBNMs can fit well (in the Appendix). We also dropped all those examples for which hair color is not unique, obtaining approx. $127k$ examples. For dSprites and MPI3D, we used a random 80/10/10 train/validation/test split, while for CelebA we kept the original split Liu et al. [2015].

We generated the ground-truth labels $y$ as follows. For dSprites, we labeled images according to a random but fixed linear separator defined over the *continuous* generative factors, chosen so as to ensure that the classes are balanced. For MPI3D and CelebA, we focused on the *categorical* factors instead. Specifically, we clustered all images using the algorithm of Huang [1997], for a total of 10 and 4 clusters for MPI3D and CelebA respectively, and then labeled all examples based on their reference cluster. This led to slightly unbalanced classes containing different percentages of examples, ranging from $5\%$ to $16\%$ in MPI3D and from $21\%$ to $29\%$ in CelebA.

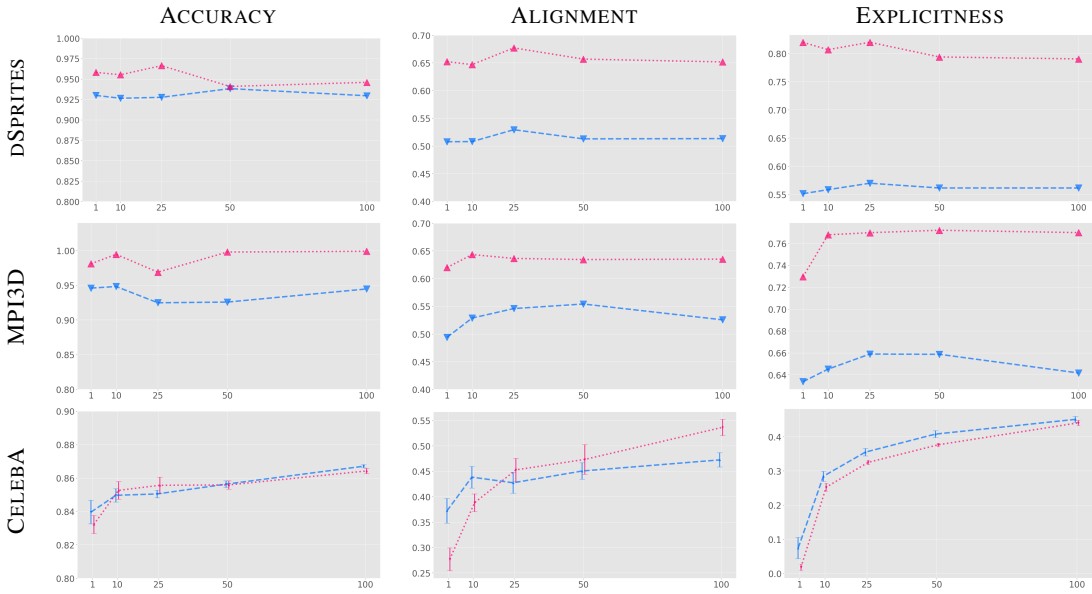

Figure 2: **GlanceNets are better aligned than CBNMs.** Each row is a data set and each column reports a different metric. The horizontal axes indicate the % of training examples for which supervision on the generative factors is provided.

**Results and discussion.** The results of this first experiment are reported in Fig. 2. All models were tested with as many latent components as the number of supervised generative factors for each dataset. The behavior of both competitors on dSprites and MPI3D was extremely stable, owing to the fact that these data sets cover an essentially exhaustive set of variations for all generative factors, so we report their hold-out performance on the test set. Since for CelebA variance was non-negligible, we ran both methods 7 times varying the random seed used to initialize the network.

The plots clearly show that, although the two methods achieve high and comparable accuracy in all settings, **GlanceNets** attain better alignment in all data sets and for all supervision regimes than **CBNMs**, with a single exception in CelebA using low values of supervision, for a total of 13 wins out of 15 cases. In all *disentanglement* data sets, there is a clear margin between the alignment achieved by GlanceNets and that of CBNMs: performances vary up to maximum of 15% in dSprites, and a minumum of 8% in MPI3D. In CelebA, the gap is evident with full supervision (almost 8% of difference in alignment), and GlanceNets still attain overall better scores in the 25% and 50% regime. On the other hand, performance are lower, but comparable, with 10% supervision. The case at 1% refers to an extreme situation where both CBNMs and GlanceNets struggle to align with generative factors, as is clear also from the very low explicitness. In dSprites and MPI3D, both GlanceNets and CBNMs quickly achieve very high alignment at 1% supervision, as expected Locatello et al. [2020], whereas better results in CelebA are obtained with growing supervision. Furthermore, both models display similar stability on this data set, as shown by the error bars in the plot.

## 5.2 EVALUATING LEAKAGE

Next, we evaluated robustness to concept leakage in two scenarios that differ in whether the unobserved generative factors are disentangled with the observed ones or not, see Section 3. In both experiments, we compare GlanceNets with a CBNM and a modified GlanceNet where the open-set recognition component has been removed (denoted CG-VAE).

**Leakage due to unobserved entangled factors.** We start by replicating the experiment of Mahinpei et al. [2021]: the goal is to discriminate between even and odd MNIST images using a latent representation $\mathbf{Z} = (Z_4, Z_5)$ obtained by trained (with complete supervision on the generative factors) *only* on examples of 4's and 5's. Leakage occurs if the learned representation can be used to predict the remaining eight digits better than random. During training, we use digit labels for conditioning the prior $p(\mathbf{Z} \mid \mathbf{Y})$ of the GlanceNet.

Fig. 3 (a, b) illustrates the latent representations of the training and test set output by a GlanceNet: since the two digits are mutually exclusive, the model has learned to map all instances along the $(z_4, z_5)$ diagonal. This is where open-set recognition kicks in: if an input is identified as open-set, the GlanceNet rejects it. In all leakage experiments, we implement rejection by predicting a random label. Since MNIST is balanced, we measure leakage by computing the difference in accuracy between the classifier and an ideal random predictor, i.e., $2 \cdot |\text{acc} - \frac{1}{2}|$: the smaller, the better. The results, shown in Fig. 3 (c), show a substantial difference between GlanceNet and the other approaches. Consistently with the values reported in [Mahinpei et al., 2021], CBNMs are affected by a considerable amount of leakage, around

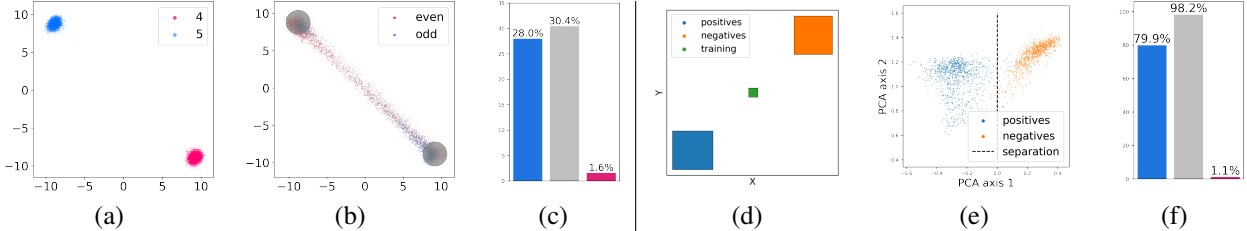

Figure 3: **GlanceNets are leak-proof.** (*a*) MNIST training set embedded using GlanceNet; axes indicate $z_4$ and $z_5$ and color the concept label (4 and 5). (*b*) Latent representations of the test images, divided in even vs. odd. Every ball in light gray denotes the region $\mathcal{Z}_y$ for each class $y$. (*c*) Leakage % for CBNM, CG-VAE and GlanceNet. (*d*) dSprites: the variations over *pos_x* and *pos_y* for the training set, and for the test set, divided in positives vs. negatives. (*e*) PCA reduction for GlanceNet. (*f*) Leakage % for CBNM, CG-VAE and GlanceNet.

28%. This is not the case for our GlanceNet: most (approx. 85%) test images are correctly identified as open-set and rejected, leading to a very low (about 2%) leakage, 26% less than CBNMs. The results for CG-VAE also indicate that removing the open-set component from GlanceNets dramatically increases leakage back to around 30%.

**Leakage due to unobserved disentangled factors.** Next, we analyze concept leakage between *disentangled* generative factors using the dSprites data set. To this end, we defined a binary classification task in which the ground-truth label depends on *position_x* and *position_y* only. In particular, instances within a fixed distance from $(0,0)$ are annotated as positive and the rest as negative, as shown in Fig. 3(a). In order to trigger leakage, all competitors are trained (using full concept-level supervision) on training images where *shape*, *size* and *rotation* vary, but *position_x* and *position_y* are almost constant (they range in a small interval around $(0.5, 0.5)$, cf. Fig. 3(d)). leakage occurs if the learned model can successfully classify test instances where *position_x* and *position_y* are no longer fixed.

For both competitors, we encode *shape* using a 3D one-hot encoding and *size* and *rotation* as continuous variables. During training, we use the *shape* annotation for conditioning the prior $p(\mathbf{Z} \mid \mathbf{Y})$ of the GlanceNet. The first two PCA components of the latent representations acquired by our GlanceNet are shown, rotated so as to be separable on the first axis, in Fig. 3 (e): in particular, it is possible to separate positives from negatives based on the obtained representations in the five latent dimensions. As shown in Fig. 3 (f), this means that both CBNM and CG-VAE suffer from very large leakage, 80% and 98%, respectively. In contrast, open-set recognition allows GlanceNet to correctly identify and reject almost all test instances, leading to negligible leakage.

## 6 RELATED WORK

**Concept-based explainability.** Concepts lie at the heart of AI [Muggleton and De Raedt, 1994] and have recently resur-

faced as a natural medium for communicating with human stakeholders [Kambhampati et al., 2022]. In explainable AI, this was first exploited by approaches like TCAV [Kim et al., 2018], which extract local concept-based explanations from black-box models using concept-level supervision to define the target concepts. Post-hoc explanations, however, are notoriously unfaithful to the model's reasoning [Sixt et al., 2020]. CBMs, including GlanceNets, avoid this issue by leveraging concept-like representations directly for computing their predictions. Existing CBMs model concepts using prototypes [Li et al., 2018, Chen et al., 2019] or other representations [Koh et al., 2020, Chen et al., 2020], but they seek interpretability using heuristics, and the quality of concepts they acquire has been called into question [Nauta et al., 2021, Margeloiu et al., 2021]. Our work shows that disentangled representation learning helps in this regard.

**Disentanglement and interpretability.** Interpretability is one of the main driving factors behind the development of disentangled representation learning [Bengio et al., 2013, Kulkarni et al., 2015, Chen et al., 2016]. These approaches however make no distinction between interpretable and non-interpretable generative factors and generally focus on properties *of the world*, like independence between causal mechanisms [Schölkopf et al., 2021] or invariances [Higgins et al., 2016]. Interpretability, however, depends on human factors that are not well understood and therefore usually ignored [Miller, 2019]. The link between disentanglement and interpretability has never been made explicit. Importantly, in contrast to alignment, disentanglement does not require that the map between matching generative and learned factors preserves semantics. Other VAE-based classifiers either do not tackle disentanglement or are unconcerned with concept leakage [Xu and Sun, 2016, Sun et al., 2020].

**Disentanglement and CBMs.** Neither the literature on disentanglement nor the one on CBMs have attempted to formalize the notion of interpretability or to establish a proper link between the latter and disentanglement. The work of Kazhdan et al. [2021] is the only one to compare disentanglement and concept acquisition, however it makes no attempt at linking the two notions. Our work fills this gap.

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

# A  IMPLEMENTATION DETAILS

## A.1  GLANCENET AND CBNM ARCHITECTURES

All experiments were implemented using Python 3 and Pytorch [Paszke et al., 2019] and run on a server with 128 CPUs, 1TiB RAM, and 8 A100 GPUs. GlanceNets were implemented on top of the `disentanglement-pytorch` [Abdi et al., 2019] library. All alignment and disentanglement metrics were computed with `disentanglement_lib` [Locatello et al., 2019] for dSprites and MPI3D. Code for the complete experimental setup is available on GitHub, and will be released upon acceptance. For each experiment, we used exactly the same architecture and number of latent variables for both GlanceNets and CBNMs to ensure a fair comparison.

**Encoder architectures:**

- *dSprites*: We chose a rather standard architecture [Abdi et al., 2019]. It comprises six 2D convolutional layers of depth 32, 32, 64, 128, 256, and 256, respectively, all with a kernel of size 4, stride 2, and padding 1, and followed by ReLU activations. The output is flattened to a vector and passed through a dense layer to obtain the mean $\boldsymbol{\mu}(\mathbf{x})$ and (diagonal) variance $\boldsymbol{\sigma}(\mathbf{x})$ of the encoder distribution $\mathcal{N}(\mathbf{Z} \mid \boldsymbol{\mu}(\mathbf{x}), \mathrm{diag}(\boldsymbol{\sigma}(\mathbf{x})))$.

- *MPI3D*: We used the same architecture with slightly different convolutional depths of 32, 32, 64, 64, 128, and 256, changing also the kernel size to 3 and removing padding, as per [Abdi et al., 2019].

- *CelebA*: We leveraged the architecture of Ghosh et al. [2020], which is a common reference for VAE models on CelebA-64 [Tolstikhin et al., 2018]. The encoder is composed of four convolutions of depth 128, 256, 512, 1024 respectively, all with kernel size of 5, stride of 2, followed batch normalization and ReLU activation.

The models had exactly as many latent variables as generative factors for which supervision is available, which in our three data sets are 7, 21, and 10, respectively.

**Decoder architecture:** All models share the same decoder architecture, obtained by stacking:

- A 2D convolution on the latent space with a filter depth of 256, kernel size of 1, and stride of 2, followed by the ReLU activation;

- Five transposed 2D convolutions of depth 256, 256, 128, 128, 64, 64, and `num_channels`, respectively, all with kernel of size 4 and stride 2.

Here, `num_channels` is either 1 (dSprites) or 3 (MPI3D and CelebA). The shape of the last layer was chosen so as to match the dimension of the input image. Additional details can be found in the various Tables in this appendix.

## A.2  SUPERVISION AND TRAINING

**Concept-level supervision.**  Depending on the supervision provided, only a fraction of the inputs was made available during training with their generative factors. In dSprites and MPI3D all generative factors are matched by the models, whereas in the case of CelebA we restricted learning to those 10 attributes that are best fit by the CBNMs, namely: `bald`, `black hair`, `brown hair`, `blonde hair`, `eyeglasses`, `gray hair`, `male`, `no beard`, `smiling`, and `wearing hat`.

**Optimization setup.**  In all experiments, we used the Adam optimizer [Kingma and Ba, 2015] with default parameters $\beta_1 = 0.9$ and $\beta_2 = 0.999$. For dSprites, we used a batch size of 64 and fixed learning rate to $\eta = 4 \cdot 10^{-4}$, while for MPI3D and CelebA we used a batch size of 100 and annealed the learning rate from $10^{-7}$ to $\eta_{MPI} = 10^{-3}$ and $\eta_{CelebA} = 10^{-4}$, respectively. To prevent overfitting, in CelebA we multiplu the learning rate by a factor of 0.95 in each epoch and apply early stopping on the validation set, with a patience of 10 epochs.

Prior to training, we selected a reasonable value for the following hyper-parameters:

- $\beta$: the weight of the KL divergence in Eq. (2).

- $\gamma$: the weight of the loss on the generative factors in Eq. (3).

- $\lambda$: the weight of the cross-entropy loss over the label, which is left implicit in Eq. (2).

Table 1: Structure of the encoder network used for dSprites.

| INPUT SHAPE | LAYER TYPE | PARAMETERS | ACTIVATION |
|---|---|---|---|
| $(64, 64, 1)$ | Convolution | depth=32, kernel=4, stride=2, padding=1 | ReLU |
| $(32, 32, 32)$ | Convolution | depth=32, kernel=4, stride=2, padding=1 | ReLU |
| $(16, 16, 32)$ | Convolution | depth=64, kernel=4, stride=2, padding=1 | ReLU |
| $(8, 8, 64)$ | Convolution | depth=128, kernel=4, stride=2, padding=1 | ReLU |
| $(4, 4, 128)$ | Convolution | depth=256, kernel= 4, stride=2, padding=1 | ReLU |
| $(2, 2, 256)$ | Convolution | depth=256, kernel=4, stride=2, padding=1 | ReLU |
| $(1, 1, 256)$ | Flatten | | |
| $(1, 256)$ | Linear | dim=7+7, bias = True | |

Table 2: Structure of the encoder network used for MPI3D.

| INPUT SHAPE | LAYER TYPE | PARAMETERS | ACTIVATION |
|---|---|---|---|
| $(64, 64, 3)$ | Convolution | depth=32, kernel=3, stride=2 | ReLU |
| $(32, 32, 32)$ | Convolution | depth=32, kernel=3, stride=2 | ReLU |
| $(16, 16, 32)$ | Convolution | depth=64, kernel=3, stride=2 | ReLU |
| $(8, 8, 64)$ | Convolution | depth=64, kernel=3, stride=2 | ReLU |
| $(4, 4, 64)$ | Convolution | depth=128, kernel= 3, stride=2 | ReLU |
| $(2, 2, 128)$ | Convolution | depth=256, kernel=3, stride=2 | ReLU |
| $(1, 1, 256)$ | Flatten | | |
| $(1, 256)$ | Linear | dim=21+21, bias = True | |

Table 3: Structure of the encoder network used for CelebA.

| INPUT SHAPE | LAYER TYPE | PARAMETERS | FILTER | ACTIVATION |
|---|---|---|---|---|
| $(64, 64, 3)$ | Convolution | depth=128, kernel=5, stride=2 | BatchNorm | ReLU |
| $(30, 30, 128)$ | Convolution | depth=256, kernel=5, stride=2 | BatchNorm | ReLU |
| $(13, 13, 256)$ | Convolution | depth=512, kernel=5, stride=2 | BatchNorm | ReLU |
| $(5, 5, 512)$ | Convolution | depth=1028, kernel=5, stride=2 | BatchNorm | ReLU |
| $(1, 1, 1028)$ | Flatten | | | |
| $(1, 1028)$ | Linear | dim=10+10, bias = True | | |

Table 4: Structure of the decoder network.

| INPUT SHAPE | LAYER TYPE | PARAMETERS | ACTIVATION |
|---|---|---|---|
| $(\dim(\mathbf{z}))$ | Unsqueeze | | |
| $(\dim(\mathbf{z}), 1, 1)$ | Convolution | depth=256, kernel=1, stride=2 | ReLU |
| $(256, 1, 1)$ | Deconvolution | depth=256, kernel=4, stride=2 | ReLU |
| $(256, 2, 2)$ | Deconvolution | depth=128, kernel=4, stride=2 | ReLU |
| $(128, 6, 6)$ | Deconvolution | depth=128, kernel=4, stride=2 | ReLU |
| $(128, 14, 14)$ | Deconvolution | depth=64, kernel=4, stride=2 | ReLU |
| $(64, 30, 30)$ | Deconvolution | depth=64, kernel=4, stride=2 | ReLU |
| $(64, 62, 62)$ | Deconvolution | depth=`num_channels`, kernel=4, stride=2 | |

For dSprites, we found a good balance for $\lambda = \gamma = 100$, while for MPI3D we achieved good performance with $\lambda = 10^3$ and $\gamma = 7 \cdot 10^3$. We adopted the same hyperparameters choice for CelebA, with the exception that we reduced the reconstruction error by 0.01. For all data sets, we cross-validated over different values of $\beta$ but we obtained better alignment performances with $\beta \approx 1$. This happens because we inject supervision on the latent factors (which is absent in regular $\beta$-VAEs Higgins et al. [2016]).

### A.3 IMPLEMENTATION OF LEAKAGE TESTS

**MNIST.** For this dataset, we considered only Multi-Latyer Perceptrons instead of convolutions. Both the encoder and the decoder are composed by two linear layers with depth 128, and a dense layer connected to the latent space and to the input space, respectively. Further details are on Table 5.

For the GlanceNet we considered a latent space of dimension 10 where the supervision on the 4 and 5 digits is used to fit the $\{z_4, z_5\}$ latent factors. These two, constitute the latent subspace where leakage occurs, while the other are useful only for reconstruction. Conversely, for the CBNM we considered only two latent factors.

During training of the latent encodings, we used stochastic gradient descent with learning rate $\eta = 0.001$, reducing it by 0.95 in each epoch for both CBNMs and GlanceNets. The training was performed only on the 4 and 5 digits (in the usual training set partition for MNIST), for almost 50 epochs. Afterwards, we considered the open-set representations, restricted to $\{z_4, z_5\}$, as inputs for training a logistic regression for parity recognition. During the training, only the digits in the MNIST training set partition (exception made for 4 and 5) are considered, while performance are calculated on the test set.

**dSprites.** We adopted the same architecture in the upper section, except that we reduced the latent space to 5 dimensions. As a reminder, during training all sprites are almost fixed at the center, therefore additional factors of variations for its position are needless. The training was performed over 300 epochs for both GlanceNets and CBNMs, with $\eta = 4 \cdot 10^{-4}$. After training, the representations of the open-set sprites (in which position is no longer fixed) are used to fit a logistic regression. In this case, the labels depend on whether the sprite is located at bottom-left corner or at the upper-right one, for more information refer to Fig. 3. The classification performance was evaluated on a held-out test set for both models, under an 80/20 train/test split.

Table 5: Encoder and Decoder structures for MNIST

| TYPE | INPUT SHAPE | LAYER TYPE | PARAMETERS | ACTIVATION |
|---|---|---|---|---|
| ENCODER | | | | |
| | $(28, 28)$ | Flatten | | |
| | $(784)$ | Linear | dim=128, bias=True | ReLU |
| | $(128)$ | Linear | dim=128, bias=True | ReLU |
| | $(128)$ | Linear | dim=10+10, bias=True | |
| DECODER | | | | |
| | $(\dim(\mathbf{z}))$ | Linear | dim=128, bias=True | ReLU |
| | $(128)$ | Linear | dim=128, bias=True | ReLU |
| | $(128)$ | Linear | dim=728, bias=True | |
| | $(728)$ | Unsqueeze | | |

# B  DCI FRAMEWORK

In our case study, we are interested into DCI maps that linearly connect the $\mathbf{g}'s$ to the $\mathbf{z}'s$. In order to evaluate alignment performances, the inverse map $\alpha^{-1} : \mathbb{R}^k \to \mathbb{R}^{n_I}$ is constructed from the latent space to the span of the $n_I$ generative factors. The latent representations and generative factors were normalized in the $[0, 1]$ interval prior to learning.

## B.1  ALIGNMENT AND EXPLICITNESS

The importance weights of this map are the absolute-values of the weights in the linear matrix of $\alpha^{-1}$, indicated as $B \in \mathbb{R}^{k \times n_I}$ in the main text, see Section 3. Then, the importance weights are used to evaluate the dispersion of the learned weights. To this end, we measure each Shannon entropy $h_j$ on all $k$ latent factors:

$$H_j = -\sum_{i \in 1}^{n_I} \bar{b}_{ji} \log_n \bar{b}_{ji} \quad \text{where} \ \ \bar{b}_{ji} = b_{ji} \Big/ \sum_{\ell=1}^{n_I} b_{j\ell}$$

Then, the average alignment is calculated as:

$$\text{alignment} = 1 - \sum_{j=1}^{k} \rho_j H_j \quad \text{where} \ \ \rho_j = \sum_{i=1}^{n_I} b_{ji} \Big/ \sum_{j'=1, i=1}^{k, n_I} b_{j'i}$$

varying from $0$ to $1$. We also calculate the explicitness of the map $\alpha$, which is related to the mean squared error (MSE) of the prediction. Since the MSE for random guessing for a variable in the $[0, 1]$ interval is equal to $1/6$, the explicitness becomes:

$$\text{explicitness} = 1 - 6 \cdot \text{MSE}$$

## B.2  EMPIRICAL EVALUATION

For dSprites and MPI3D, all DCI quantities were calculated with the built-in evaluation code provided by `disentanglement_lib`, Locatello et al. [2019]. For CelebA, since the 40 attributes in CelebA are not exhaustive for the image generation, we implemented computed DCI as follows: *(i)* we first converted the $J$ attributes $\mathbf{z}_J$ and $\mathbf{g}_J$ connected to `hair type` to a single concept $h$ and fit the model with Lasso regression to predict $g_h$ from $\mathbf{z}$. Then, *(ii)* we trained a Logistic Regression with $l1$ penalty to predict the remaining $\mathbf{g}'s$. Finally, we took both weights in *(i)* and in *(ii)* to compute the matrix $B \in \mathbb{R}^{6 \times 6}$. In this way, we determined alignment and explicitness for CelebA. We chose the lasso coefficient $\lambda = 0.01$ for both regressions.

# C QUALITATIVE RESULTS FOR CONCEPT LEAKAGE IN MNIST

In Fig. 4, we show the latent space representations for different models on the MNIST leakage test, for both closed-set and open-set data points. To illustrate the contribution of our mixture prior, in addition to the CBNM and GlanceNet models, we also considered a simpler supervised VAE model. This model has the same encoder, decoder, and classifier as the GlanceNet, but uses a regular Gaussian prior[2]. We found that this model achieved a similar level of leakage to CG-VAE. We display in Fig. 5 the reconstruction of a few random examples output by GlanceNet: the reconstructions of all instances belonging to the open-set greatly deviate from the original.

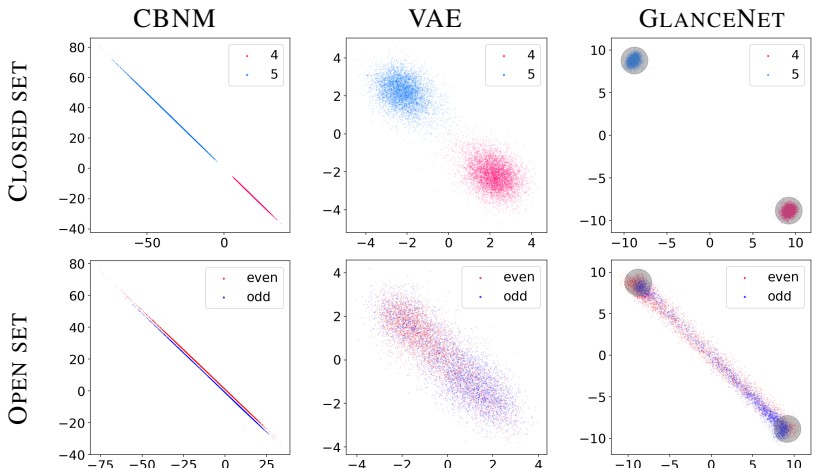

Figure 4: **Latent space representation for MNIST.** On the first row, we report the representations for 4 and 5 as fitted by CBNM, VAE and GlanceNet, respectively. On the second row, we display the scattering plot for points only belonging to the open set. For CBNM, we separated even and odd instances by $\Delta y = 2$, since their representations strongly overlap. All plots comprise only the $z_4, z_5$ axes.

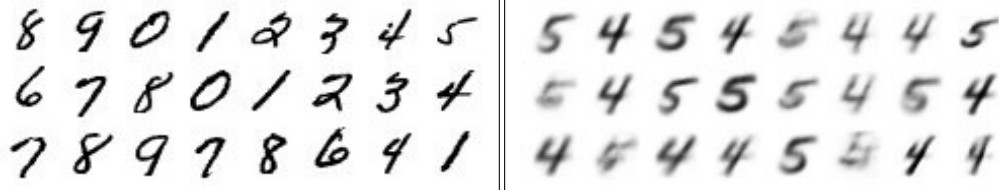

Figure 5: **MNIST reconstruction with GlanceNet.** On the left we reported the original digits, whereas on the right the reconstruction with the learned decoder. All images have been inverted in the black and white scale.

---

[2]For the VAE model, we chose the Gaussian prior in Kingma and Welling [2014], i.e., $p(\mathbf{z}) = \mathcal{N}(\mathbf{z}|0, 1)$.

# D   QUALITATIVE RESULTS FOR CONCEPT LEAKAGE ON DSPRITES

We also include qualitative results for GlanceNet and on dSprites for closed set and open set data points. In Fig. 6 we display the projections of train and test points on the two different latent subspaces (see caption). In both of them, positives and negatives representations are well separated from each other, implying substantia leakage. We also evaluated the reconstruction quality during training and testing and reported some of them in Fig. 7. Notably, almost all points are recognized to be open set instances thanks to the reconstruction threshold.

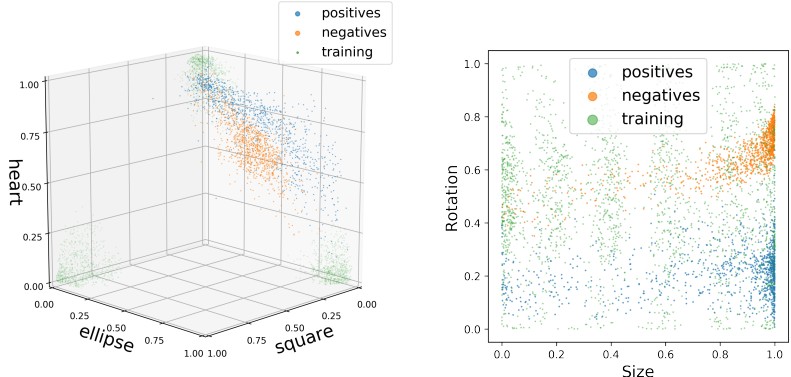

Figure 6: **Concept space representation of GlanceNet for dSprites.** On the left, we show the projections on the one-hot encoded `shape` subspace, whereas on the right we project on the {`size`, `rotation`} subspace. We include the representations for training points, positive and negative ones.

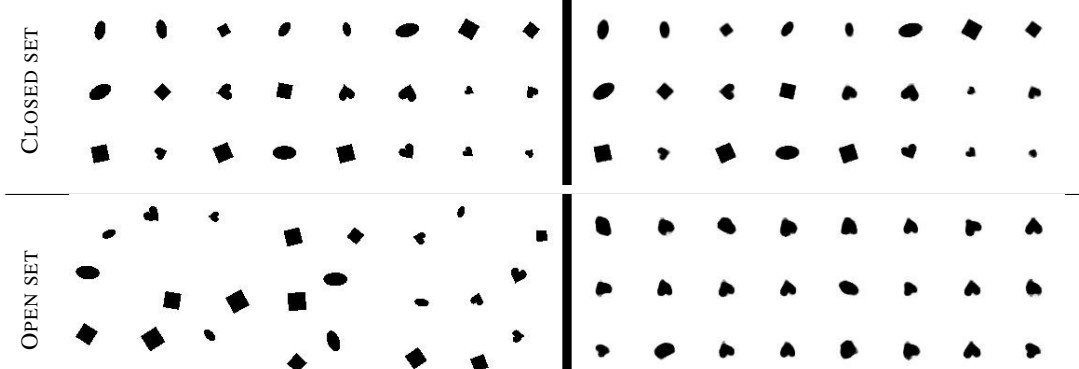

Figure 7: **Reconstruction for dSprites on train and test with GlanceNet.** On the upper panel, we report the reconstructions of the sprites belonging to the closed set. On the lower one, the reconstructions of the open set points. Like MNIST, all images have been inverted in the black and white scale.

