# OpenReview forum: "GlanceNets: Interpretabile, Leak-proof Concept-based Models"
_auai.org/UAI/2022/Workshop/CRL — CRL@UAI 2022 Poster_

### Official Review · Reviewer_t9VV · 2022-06-28
**Review: GlanceNets**

**Rating:** 7
**Confidence:** 3

**Review:**

### Summary:
This paper aims to address two deficiencies in concept-based models (CBMs), namely: a lack of interpretability and concept leakage. To address the lack of interpretability, the authors propose to realise the interpretable concepts as the ground-truth latent factors in the data for which some supervision is assumed. To address concept leakage, the authors use a method called open-set recognition aimed at identifying the OOD inputs which are purported to cause concept leakage. The author’s introduce a VAE-based model called GlanceNets integrating both of these ideas which is shown to outperform a baseline in terms of interpretability and robustness to concept leakage.

### Strengths of Paper:
* The paper is well written and the contributions are presented clearly and concisely.
* I found the overall idea of connecting the interpretable concepts in a CBM with disentangling the ground-truth latent factors to be a step in the right direction for formalising the slippery notion of interpretability.
* Recognising only a subset of the generative factors as being interpretable I think is the right assumption to make and is often overlooked in the disentanglement literature.
* The experimental results appear promising. In particular, the improvements over the CBNM baseline in the alignment experiments seem particularly interesting and non-trivial as my understanding is that these models are trained directly on the latent concepts.

### Areas for Improvement:
* The authors state that there is a distinction between their proposed definition of “alignment” and that of identifiability. It’s not clear to me, however, how this is the case. Effectively, what the author’s are proposing as “alignment" is identifiability up to permutation and point-wise transformations of the latents which is in line with notions of identifiability pursued in the literature (see, e.g. [1,2]).
*  It was not clear to me from reading the main text exactly how the CBNM models were implemented. Thus, while the empirical results of GlanceNets over CBNMs seems promising, I found it difficult to assess the significance definitively in the absence of a clearer description of CBNMs.
* The authors make a point in the main text to state that only a subset of the generative factors are assumed to admit a semantic interpretation, however, in the alignment experiments, the latent dimension of their models was set to the number of supervised generative factors available. This seems to defeat the purpose of assuming this partitioning of the latent space into semantic and non-semantic latents.
* In figure 2, a legend stating what the the different colors represent should be placed somewhere and not just in the main text.
* In Figure 3, it would be useful if the axis for panels a. and b. were labelled.

### Conclusion:
Overall, I think this paper takes a step in an important direction of leveraging disentangled representation learning to build more interpretable classifiers. The contributions are presented clearly and the empirical results are promising, thus I would recommend acceptance. In future work, it would be interesting to explore how more coarse-grained latent concepts could be integrated into CBMs such as objects which are best represented via multiple latent factors opposed to a single latent dimension.

### References:
[1] https://arxiv.org/abs/1907.04809

[2] https://arxiv.org/abs/2002.11537

---

### Official Review · Reviewer_t726 · 2022-06-30
**Good experiments showing better alignment and less leakage of a new CBM class**

**Rating:** 7
**Confidence:** 3

**Review:**

**Summary of the Paper**
This paper introduces a new class of concept-based models (CBMs) called GlanceNets with the motivation of learning interpretable representations that are robust to concept leakage. The paper also proposes a definition for interpretability of a learnt representation as its alignment with the user’s representation, and a notion of concept leakage based on domain shift. The proposed GlanceNets are based on the idea of using the classifier for open-set prediction on top of a $\beta$-VAE. GlanceNets are evaluated on classification tasks based on the dSprites, MPI3D, and CelebA datasets for measuring alignment, and on MNIST for measuring how league-proof they are.


**Main Review**

*Contributions*

The proposed GlanceNets are shown to have greater alignment and accuracy (except on CelebA, nearly similar) than concept bottleneck models (CBNMs) on all the datasets considered (as seen in Figure 2) showing GlanceNets can learn a useful map between generative and latent factors.
GlanceNets are shown to have lower leakage than CBNMs and CG-VAEs by performing open-set prediction and rejecting the inputs identified as open-set, also shown qualitatively for the learned latent space representations.
The authors propose a definition of interpretability based on alignment between the generative factors $G$ of the DGP and the learned latent factors $Z$ such that the mapping between the two doesn’t mix multiple $G$s into a single $Z$.
The authors also propose a notion of concept leakage based on domain shift of the generative factors.

*Quality*

The quality of the paper is good, the figures are helpful.

*Clarity*

The paper is decently well-written though some parts are a bit difficult to fully appreciate–
- The connection made between disentanglement and interpretability or CBMs is not very clear– I think the connection between alignment and disentanglement can be more clearly written.
- The part on interpretability and concept leakage is not very clear– how exactly does domain shift come into the picture, and how does all this relate to OOD generalization?
- It would be useful to have a legend with the plots (especially the ones in  Figure 2) apart from the colour-coding in the text.

*Strengths*
- Experiments are well-explained and details about the architectures and hyperparameters are provided.

*Weaknesses*
- The proposal of connecting disentanglement and interpretability is interesting, but could be more well-defined– why do we want to preserve semantics, and how does it exactly connect with concept leakage?
- The paper would benefit from an algorithm/pseudocode laying out an overview of all the entire process.

**Originality and Significance**

The work looks novel as compared to previous work on CBMs. However, I am a bit unclear on how significantly it fits into the big picture– I think the current work can greatly benefit from making more substantiated connections between disentanglement and interpretability– which is something the authors propose to do, but in my opinion, doesn’t come out that well.

---

### Meta-Review · Program_Chairs · 2022-07-06

**Recommendation:** Accept (Poster)
**Confidence:** 4

**Metareview:**

This is a good paper and should be accepted at the workshop.

---

### Decision · Program_Chairs · 2022-07-06

Accept (Poster)